ecology/behaviour/plant science

bryophyte dispersal, epizoochory, bipartite network, avian–bryophyte interactions, dispersal ecology

**Author for correspondence:**
Matthew W. Chmielewski
e-mail: mwchmielewski@gmail.com

# Species-specific interactions in avian–bryophyte dispersal networks

## Matthew W. Chmielewski and Sarah M. Eppley

Department of Biology, Portland State University, 1719 SW 10th Avenue, SRTC rm 246, Portland, OR 97201, USA

MWC, 0000-0002-4467-4977

Studies from seed plants have shown that animal dispersal fundamentally alters the success of plant dispersal, shaping community composition through time. Our understanding of this phenomenon in spore plants is comparatively limited. Though little is known about species-specific dispersal relationships between passerine birds and bryophytes, birds are particularly attractive as a potential bryophyte dispersal vector given their highly vagile nature as well as their association with bryophytes when foraging and building nests. We captured birds in Gifford Pinchot National Forest to sample their legs and tails for bryophyte propagules. We found 24 bryophyte species across 34 bird species. We examined the level of interaction specificity: (i) within the overall network to assess community level patterns; and (ii) at the plant species level to determine the effect of bird behaviour on network structure. We found that avian–bryophyte associations are constrained within the network, with species-specific and foraging guild effects on the variety of bryophytes found on bird species. Our findings suggest that diffuse bird–bryophyte dispersal networks are likely to be common in habitats where birds readily encounter bryophytes and that further work aimed at understanding individual bird–bryophyte species relationships may prove valuable in determining nuance within this newly described dispersal mechanism.

## 1. Background

Dispersal is often a brief period of an organism's life history but is an especially important life stage for sessile organisms such as plants, which otherwise maintain a sedentary lifestyle. Dispersal sets the context for the majority of the plant life cycle, from germination through to senescence. By moving, plant propagules are potentially able to escape competition with conspecifics and avoid density-dependent mortality owing to herbivores and/or parasites [1,2]. The deposition of propagules

in locations with particular characteristics can largely impact seedling germination and survival [3,4]. At a population level, dispersal maintains gene flow within metapopulations and increases connectivity across the landscape [5]. In addition, it plays a pivotal role in ecological assembly by generating and maintaining diversity within and among communities [6,7]. The movement of propagules can drive community composition through priority effects [8,9] as well as the differential species-specific dispersal of seeds [10]. Dispersal contributes to inter-community connectivity [11,12], and diversity within and among communities depends on the extent of connectivity among parts of a metacommunity [13] and varies across spatial scale [14].

Mechanisms that increase the probability of a plant settling in a particular locale within the landscape can have outsized impacts on the distribution of a wide variety of taxa and the subsequent local community composition [15]. Biased movement towards appropriate habitat can improve the quality of dispersal outcomes, increasing the effectiveness of the dispersal event [16]. Animal vectors are particularly suited to altering plant propagule distribution patterns by integrating behaviourally directed propagule deposition towards appropriate locations within a largely heterogenous environment [17–19]. Indeed, behaviourally mediated dispersal via associations with animal vectors has been shown to have widespread impacts on plant propagule outcomes [20,21]. Birds, being highly vagile organisms that are sensitive to particular habitat characteristics, are especially likely to increase dispersal distance for vascular plants as well as improve quality and connectivity of seed dispersal from local to landscape scales [22–26].

While much effort has been applied towards understanding how animal behaviour may shape the dispersal outcomes of seed plants, we know little about the ramifications of animal dispersal of plant spores. Plant reproduction via spores predates the evolution of seeds, and understanding more about the dispersal strategies of spore-bearing plants may provide insight into the context in which seeds and seed dispersal evolved [27–29].

Researchers have argued that spore-bearing plants are unlikely to be limited by dispersal, but studies have been mainly focused on *Sphagnum* spp. [30,31], inferring dispersal distance [32,33] or implying that niche constraints solely drive bryophyte distributions [34]. Mounting evidence suggests that dispersal can impact richness, diversity and metacommunity dynamics in bryophytes [35–39]. Multiple studies have shown that mammals and invertebrates harbour bryophyte diaspores and may therefore be contributing to bryophyte dispersal outcomes [40–44]. Recently, mounting evidence has suggested that birds may also be effective bryophyte dispersers. Studies have shown that bryophytes can survive the gut passage of water birds, suggesting that some cases of endozoochory may occur [45]. The hummingbird *Sephanoides sephaniodes* has been found to transport gametophytic material between sites to form nests (synzoochory), and these bryophytes were shown to be capable of reproducing in their new sites [46]. Furthermore, Fontúrbel and colleagues [46] have found that *S. sephaniodes* is selective in choosing bryophytes as nest material and suggest that these bryophyte species thereby play a central role in tripartite hummingbird bryophyte tree networks. Lewis *et al.* [47] found bryophyte propagules on nesting shorebirds, and while dependent on a small sample, this work provided the first published evidence of naturally occurring bryophyte propagules on bird surfaces (epizoochory). In additional work examining the intercontinental structure of populations of *Tetraplodon*, Lewis *et al.* [48] found evidence for direct long distance dispersal between disjunct high latitude locations between hemispheres, which they suggest may be possibly attributed to dispersal by birds. Our own work has found that bryophyte spores are found on the feathers and legs of a wide range of passerine taxa and that these spores are viable [49]. While these studies provide preliminary suggestions that birds may transport bryophyte propagules, further work that considers species-specific dispersal interactions is necessary to understand the potential importance and underlying structure of this dispersal mechanism.

To determine whether avian species and foraging behaviour influence the diversity and identity of topically vectored bryophyte species, we captured passerine birds within Gifford Pinchot National Forest, WA, Pacific Northwest USA and sampled them for bryophyte propagules. Passerines are the most diverse and widespread group of birds, performing vast migration each year with stopovers that may link distant patches of habitat that other modes of dispersal would otherwise be unlikely to connect [50,51]. We sampled bird legs and tail feathers, germinated samples and used a chloroplast marker to identify individual bryophyte species associated with each bird. Epiphytic cryptogams exhibit vertical stratification within forests [52,53], including at the Wind River Experimental Forest, directly adjacent to our site [54,55] where bryophyte diversity is known to decrease with substrate height. We therefore predicted that: (i) bird species would vary in the diversity of bryophyte propagules that they carried owing to differential use of habitat; (ii) that specialization of associations

of individual bryophyte species would increase with increasing substrate foraging height; and (iii) the diversity of bryophytes found on a focal avian group (across both species and behavioural guild) would be constrained relative to a random sample from the available species pool.

# 2. Methods

## 2.1. Mist netting and sampling for bryophytes

To sample bird surfaces for bryophyte propagules, we captured birds along a recreational trail adjacent to the Wind River Experimental Forest within Gifford Pinchot National Forest, WA, USA. We deployed ten $12 \times 3$ m, 30 mm mesh mist nets from dawn to midafternoon throughout an Oregon ash (*Fraxinus latifolia* Bentham) forest, surrounding a banding station at 45 48′40″ N, 121 56′35″ W. We checked nets at least every 30 min, retaining birds for banding and sampling prior to release. Our site is adjacent to mixed western hemlock (*Tsuga heterophylla* Sargent) and Douglas fir (*Pseudotsuga menziesii* (Mirbel)) forest and is bounded on one side by a small patch of meadow. The variety of surrounding habitat at our field site generates a diversity of avifauna with different habitat and foraging predilections. We sampled the legs and tails of each captured individual for bryophyte propagules using cotton swabs (see [49] for details). Birds were additionally fitted with United States Geological Survey (USGS) identification leg bands to account for recaptures. Avian capture data, along with age, sex and morphometrics, were submitted the to the USGS Bird Banding Laboratory. In order to better contextualize the bryophytes sampled from bird surfaces relative to the abundance of bryophytes in our field site, we measured the cover of bryophytes on both tree trunks and the ground by quadrat sampling every 10 m along multiple transects. We sampled along transects both in the Oregon ash dominant forest that contained our mist nets, as well as adjacent Douglas fir dominant forest.

## 2.2. Sample processing and molecular methods

Samples were vortexed twice each for 1 min in filtered tap water and vacuum filtered onto gridded 0.45 μm mixed cellulose ester membranes (EMD Millipore). We placed filters onto $60 \times 15$ mm Petri plates containing BCD nutrient agar [56] and grew them under a $12:12$ L : D light cycle at approximately 500 lux at room temperature (22–25°C). Plates that germinated over the next year were stored and later sampled for tissue. DNA was extracted using the manufacturer's protocol (Sigma Aldrich Extract-N-Amp PCR kit) and frozen for storage. Samples were later thawed, and the trnF-L region [57–60] of the chloroplast genome was amplified via polymerase chain reaction (PCR) using the manufacturer's protocol. PCR products were Sanger sequenced on ABI $3730\times1$ instruments (Functional Biosciences, Inc.). Additionally, we sequenced known species collected at our field site in order to improve our ability to confirm species calls.

## 2.3. Sequence processing and tree building

We aligned forward and reverse Sanger reads to generate a consensus sequence for each sample, which we then trimmed to remove primer annealing sites. Samples were then preliminarily identified by comparison with National Center for Biotechnology Information (NCBI)-accessioned trnF-L sequences via Basic Local Alignment Search Tool (BLAST) search [61]. Both bird swab and known field sample sequences were aligned in GENEIOUS (v. 8.0.5) and used to build an unweighted pair group method with arithmetic mean (UPGMA) tree using the *ape* and *phangorn* packages in the R statistical computing platform v. 3.3.3 (electronic supplementary material, figure S1) [62–64]. Known samples from the field were used to confirm avian-derived sample identities before being removed from the tree. One *Orthotrichum* grouping of sequences failed to align appropriately within our tree and was removed from our analysis of phylogenetic distance (PD). The single *Marchantia* sample in our dataset was determined to be an overly influential outgroup and was removed prior to computing PD, resulting in our final tree. Both avian species and family were plotted across bryophyte species trees in order to compute PD and individual species associations were plotted across the bryophyte phylogeny.

## 2.4. Data analysis

We quantified avian–bryophyte dispersal networks by applying both bipartite network and PD approaches. We examined the specificity of interactions within the bipartite network via two commonly used indices, $H_2'$ which quantifies the level of interaction specificity within a network and $d'$ which quantifies the specificity of interactions of individual species within the network. Originally presented as network- and species-level interaction specialization [65], these indices have been variously referred to as selectivity [66], exclusiveness and specificity [67]. Some of these terms imply behaviourally directed use of resources that involve highly coevolved systems, reflecting the plant–pollinator systems that they were initially used to describe. Despite this, these indices have been used to describe the relationships within various types of interaction networks including seed dispersal syndromes [68,69], tree chemical and insect chemical networks [67], mite-microhabitat associations [70], epiphyte-phorophyte commensalisms [71] and arbuscular mycorrhizal fungi-plant associations [72]. Conceptually, we propose a framework in which bird use of particular Eltonian niche-space within the environment leads to differential exposure to bryophyte propagules. We therefore use the term interaction specificity throughout to describe the patterns of partner diversity within our study to disambiguate from other textual interpretations associated with specialization or selectivity.

In order to assess the role that foraging behaviour plays in shaping the suite of bryophyte propagules found on bird surfaces, we assigned bird species to foraging guilds based on the EltonTraits 1.0 database [73]. Species were assigned to foraging guilds (ground, understory, midhigh and/or canopy foraging) if the foraging strategy represented at least 20% of their foraging activity within the database (electronic supplementary material, figure S1).

We determined the level of interaction specificity ($H_2'$) of the overall network as well as foraging guild-based subset networks and compared observed values with a permutationally generated null distribution of values [74]. Each null distribution was generated by first generating a distribution of marginal values from the focal interaction matrix, followed by sampling from these distributions to create new marginals and interaction networks based on the abundance cross-product of each permutation (via the null.distr function in the package bipartite) [75]. For each interaction matrix, a null distribution of 1000 permutations was generated. The significance (*p*-value) of a given network-level degree of specificity when compared to a random distribution of associations can be derived as the proportion of values of the random distribution of values that exceed or are equal to the observed $H_2'$ [65]. While some bipartite network indices are sensitive to sampling intensity, null model comparisons have been shown to be an appropriate way to account for these sensitivities [76]. In order to compare the magnitude of the effect of bird foraging strategy on $H_2'$, we also calculated $\Delta H_2'$ (observed $H_2'$ – mean null $H_2'$) [77].

In addition to network-level analyses, we also calculated species-level specificity ($d'$) of the bryophytes in the overall avian-association network, as well as subset networks defined by avian behavioural guild [78]. We assessed how avian foraging guild impacts bryophyte interaction specificity by building linear models comparing $d'$ for each bryophyte species in the total dataset to $d'$ within networks constructed from individual behavioural types.

We analysed Faith's PD by treating each avian species as an aggregate 'site', calculating the minimal path of connectivity across the bryophyte tree of all species found on each individual bird species (electronic supplementary material, figure S2). The observed PD for each species was subjected to a permutation test in which a sample-size controlled null distribution was generated by permuting bryophyte species identities randomly 1000 times across the tree and calculating minimal path of connectivity [79]. Observed values of PD were compared with the mean ± 95% confidence interval (CI) of the null distributions in order to determine whether avian-vectored bryophyte species were more clumped than random. Analyses and visualizations were constructed in R using the *bipartite*, *picante* and *ggplot2* packages [75,80,81].

# 3. Results

## 3.1. Bird and bryophyte species

During our study period, we captured 34 different species of birds comprised of 192 individuals. We most commonly captured Swainson's thrushes (*Cathartus ustulatus*, 47 captures), with rufous hummingbirds (*Selasphorus rufus*, 25), MacGillivray's warblers (*Geothlypis tolmiei*, 19), dark-eyed juncos

(*Junco hyemalis*, 13), and lazuli buntings (*Passerina amoena*, 12) being well represented in the dataset. On these 192 birds, we found 24 species of bryophytes. The most common species was *Ceratodon purpureus*, a weedy species with a wide distribution that grows on a wide variety of substrates including soils, rocks and anthropogenically disturbed substrates [82]. Other common species included *Aulacomnium androgynum*, *Isothecum myosuroides* and *Kindbergia oregana*. Similarly, our field survey of bryophytes included abundant cover of *I. myosuroides* on trees and *K. oregana* on the ground but also included abundant *Hypnum circinale* and *Neckera douglasii* which were not found on our avian samples (electronic supplementary material, figure S3).

## 3.2. Description of network and network-level analysis

The overall bird–bryophyte dispersal network exhibited non-random species-specific association structure, providing evidence that bird species identity impacts the types of bryophytes being carried on bird surfaces (figure 1; electronic supplementary material, figure S4). While some common bryophytes such as *A. androgynum* and *I. myosuroides* were found most abundantly on the commonly captured Swainson's thrush, other common mosses such as *C. purpureus* and *Racomitrium elongatum* were associated most strongly with less common birds in the network like American robins and chipping sparrows. Similarly, a variety of less common bird species at the network periphery such as hermit warblers were connected to the network via relatively uncommon bryophyte species, rather than bryophytes at the network core.

Quantitatively, the overall interaction network was more highly specialized than expected when compared with the null model distribution ($\Delta H_2' = 0.13$, $p < 0.01$; figure 2). When interaction networks of specific behavioural groups were examined, ground foragers exhibited a significant, but weak level of interaction specificity ($\Delta H_2' = 0.08$, $p < 0.01$; figure 2), while both understory ($\Delta H_2' = 0.10$, $p < 0.01$) and midhigh ($\Delta H_2' = 0.28$, $p < 0.01$) foraging birds had moderately specific associations with bryophytes. By contrast, canopy foraging birds were randomly associated with bryophyte networks ($\Delta H_2' = 0.16$, $p = 0.22$).

## 3.3. Species-level analysis

Individual bryophyte species varied in their specificity (d′) within the overall interaction network (figure 3a). When these species associations were examined within subset networks defined by avian foraging behaviour, bryophytes associating with ground foragers significantly reflected overall specificity within the network, ($F_{1,21} = 78.73$, adjR $= 0.78$, $p < 0.01$; figure 3b). Two species notably deviated from this pattern. *Polytrichum juniperinum* decreased associate specificity, while *Orthotrichum consimile* associations become more generalized. Bryophytes associated with both understory foragers were even more strongly influential in the overall network ($F_{1,20} = 116.2$, adjR$^2 = 0.85$, $p < 0.01$; figure 3c), with *O. consimile* again decreasing association specificity. Neither midhigh foraging associated bryophytes ($F_{1,12} = 3.77$, adjR$^2 = 0.18$, $p = 0.08$) nor canopy foraging associated bryophytes ($F_{1,7} = 0.03$, adjR$^2 = -0.14$, $p = 0.86$) significantly predicted overall network patterns. The PD of the bryophytes found a given bird taxon, our other measure of bryophyte–avian interaction specificity, was lower than expected by chance across the avian species we sampled (figure 4a). While the PD was highest in some species that were captured frequently (e.g. Swainson's thrushes, American robins) as would be expected, species comprising similar proportions of the overall network (e.g. hermit warblers and Pacific wrens) varied in the diversity of their bryophyte associates. Overall, phylogenetic diversity was constrained compared to the available pool of associates. At the family level, we similarly found a lower PD than expected by chance across all groups (figure 4b).

## 4. Discussion

Our results describe, to our knowledge, the first study of a bryophyte–bird interaction network that explicitly links multiple bird species to the bryophyte species carried on their surfaces (epizoochory), and to our knowledge is a first examination of these relationships in passerine birds. The variety of bryophyte propagules that we found on a wide array of passerine species suggests that this is a general phenomenon, with potential implications for understanding the impact of animal behaviour on bryophyte dispersal. Our study system reflects a network structure in which certain bryophyte–bird associations are more common than others, and this demonstrates that bipartite species

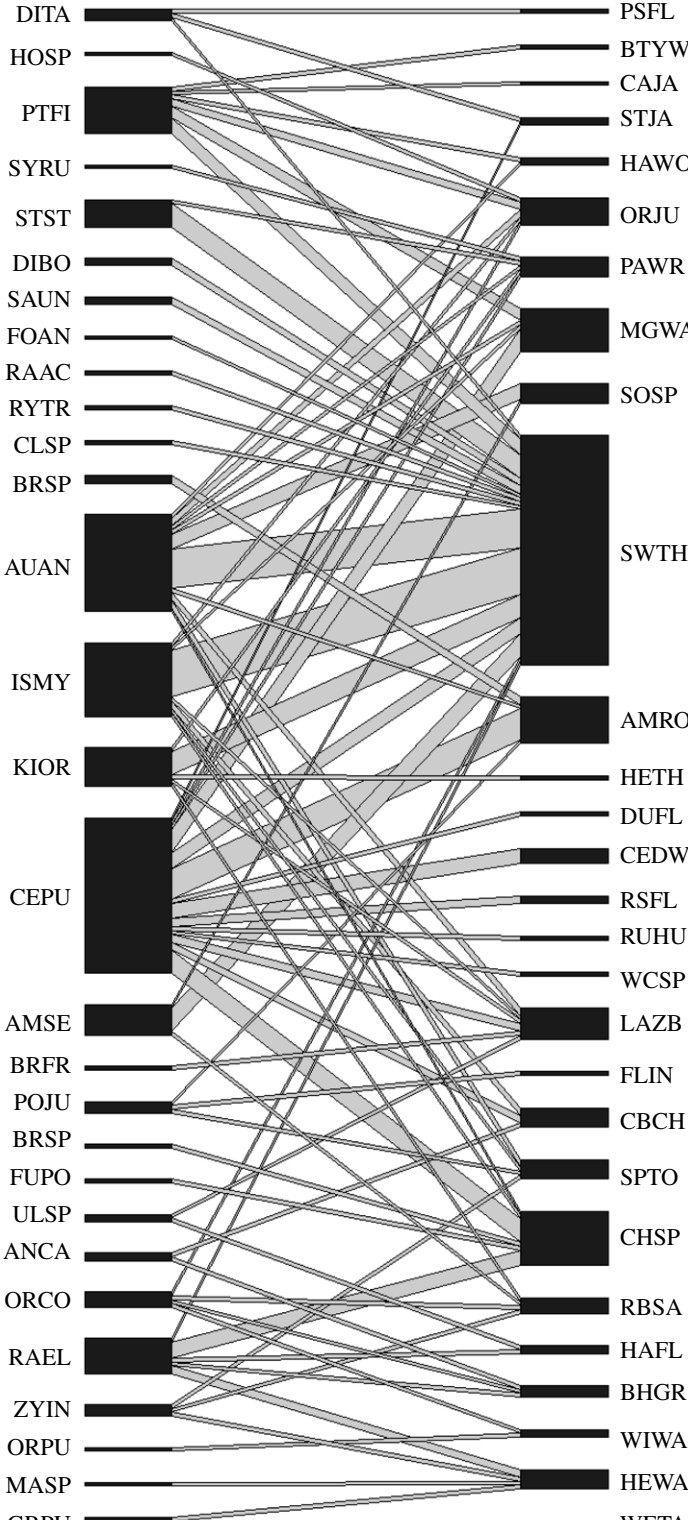

**Figure 1.** Network connectivity between bryophytes (left) and birds (right) within a dispersal association network. The relative width of species bars on either end of the association network represents the relative representation of a particular species within the dataset, with the width of connections between species representing the strength of the dispersal association. Species are identified via their Bird Banding Laboratory four-letter identification code and bryophytes by the first two letters of their genus and species names. Full names for each species are provided in the electronic supplementary material.

relationships show significant interaction specificity. While we predicted that interaction specificity in avian–bryophyte interaction networks would increase with increasing bird foraging height, our results suggest that midhigh foraging was associated with the highest level of interaction specificity at the

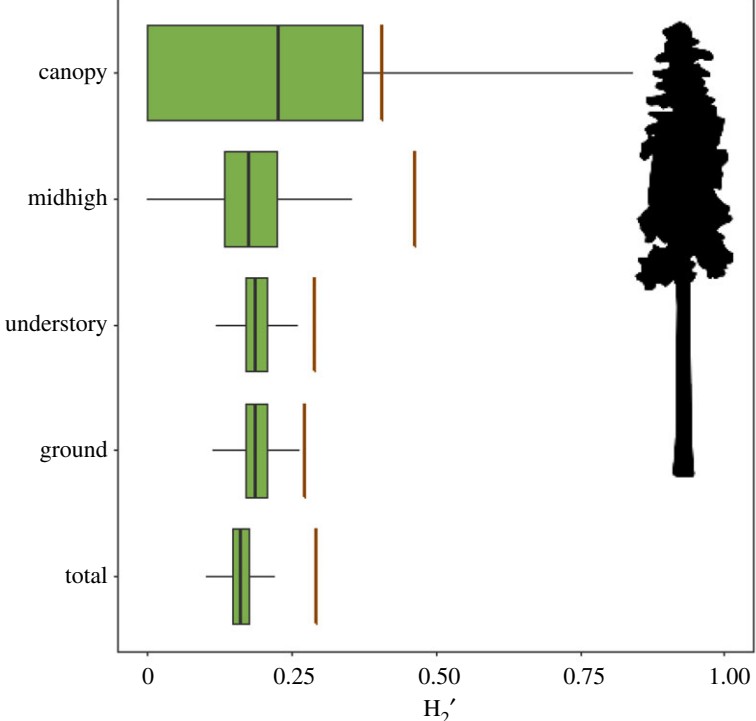

**Figure 2.** The level of observed specificity of connections within an avian–bryophyte dispersal network and subset networks based on avian foraging guild. Observed values for each network (brown vertical bars) were compared with the level of specificity of a null distribution of networks (boxplots). Higher values of $H_2'$ indicate a higher degree of specificity of network interactions within the network. Null distributions were computed by randomizing bipartite associations across the network over 1000 permutations. Silhouette is from PhyloPic.org, contributed by Michele M. Tobias.

network level, followed moderately by the understory foraging network. The weak specificity displayed by the ground foraging network may be owing to the ubiquity of bryophytes on the forest floor in our site, relative to other foraging heights. Both ground and understory networks exhibited similar behaviour, suggesting that in Pacific Northwest forests, both ground and foliage gleaning species are exposed to similar bryophyte diaspores. While these foraging guilds share many species, a number of species are distinct, but bryophyte abundance in the understory and ground may lead to homogenization of available diaspore material in these microhabitats. The random structure exhibited by canopy foraging associations may be owing to bryophyte abundance being limited in forest canopies. Together, these vertical distribution patterns may drive mid-story foraging strategies to lead to the most specialized avian–bryophyte associations.

From a bryophyte species-specific point of view, specificity varied widely, with some species (such as *Grimmia pulvinata* and *Orthotrichum pulchellum*) exhibiting tight associations with certain avian species while others (such as *Sanionia uncinata* and *Rhytidiadelphus triquetrus*) being found across a wide variety of birds. Some species were more widely distributed in particular foraging zones, with *O. consimile* exhibiting generalization in its association with both ground and understory foraging birds. Conversely, *P. juniperinum* showed tighter associations with ground foraging species compared with the rest of the species in our network, suggesting that species-specific ground foraging behaviour may be important in exposing individual birds to this moss. Contrary to our prediction that individual species specificity would increase with increasing substrate height, bryophyte associate specificity was highest at midhigh levels within forest, comparatively decreasing within the canopy. Despite the overall patterns of association specificity, the high variability of species-specific associations with birds of various foraging guilds suggests additional work will be necessary in order to better understand the role that birds may play in dispersing particular species of bryophytes.

While demonstrating a level of specificity greater than expected should bryophyte propagules be ubiquitously dispersed, the bryophytes and birds in our study system fail to exhibit tight individual species–species dependencies. This diffuse dispersal network is similar in nature to what is seen in seed plant–animal dispersal relationships, where association between plant seeds and suites of

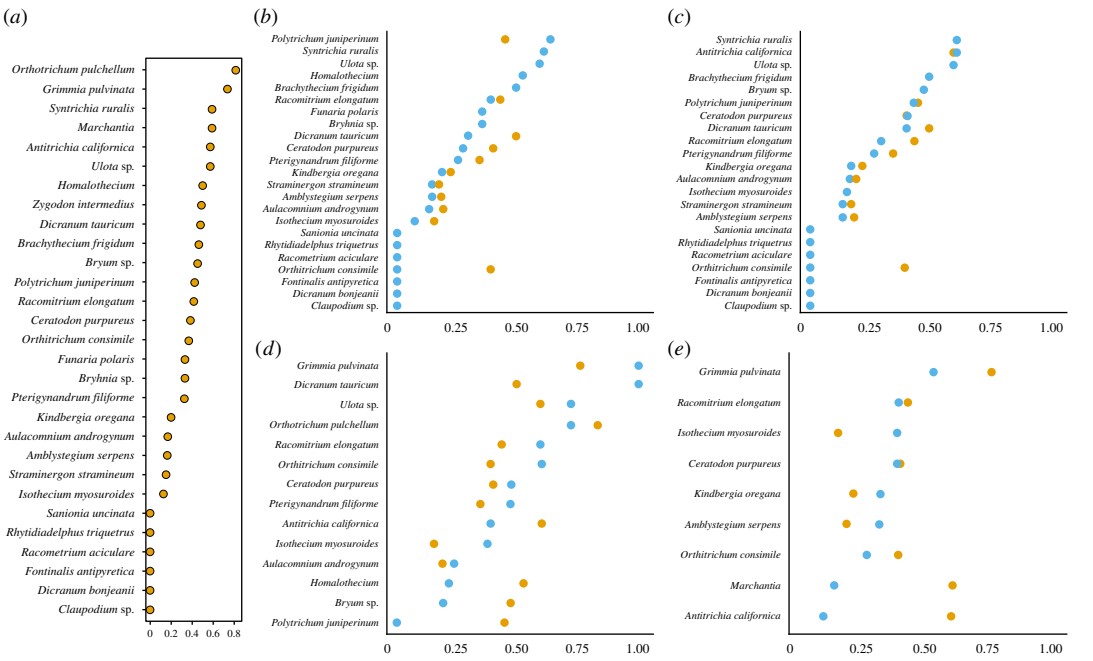

**Figure 3.** The level of specificity of individual bryophytes in an avian–bryophyte dispersal network. Higher values of d′ indicate that the individual bryophyte species has more specialized associations with birds within the bipartite network. Orange points represent the value of d′ in the overall network, while blue points represent the value of d′ in foraging-specific subset networks. Shifts to right in subset networks indicate more specific relationships within that foraging network (fewer interaction partners), while shifts to left indicate less specificity within the subset network. Species-level interaction specificity of individual bryophytes within the (*a*) overall network, (*b*) ground foraging network, (*c*) understory foraging network, (*d*) midhigh foraging network, and (*e*) canopy foraging network.

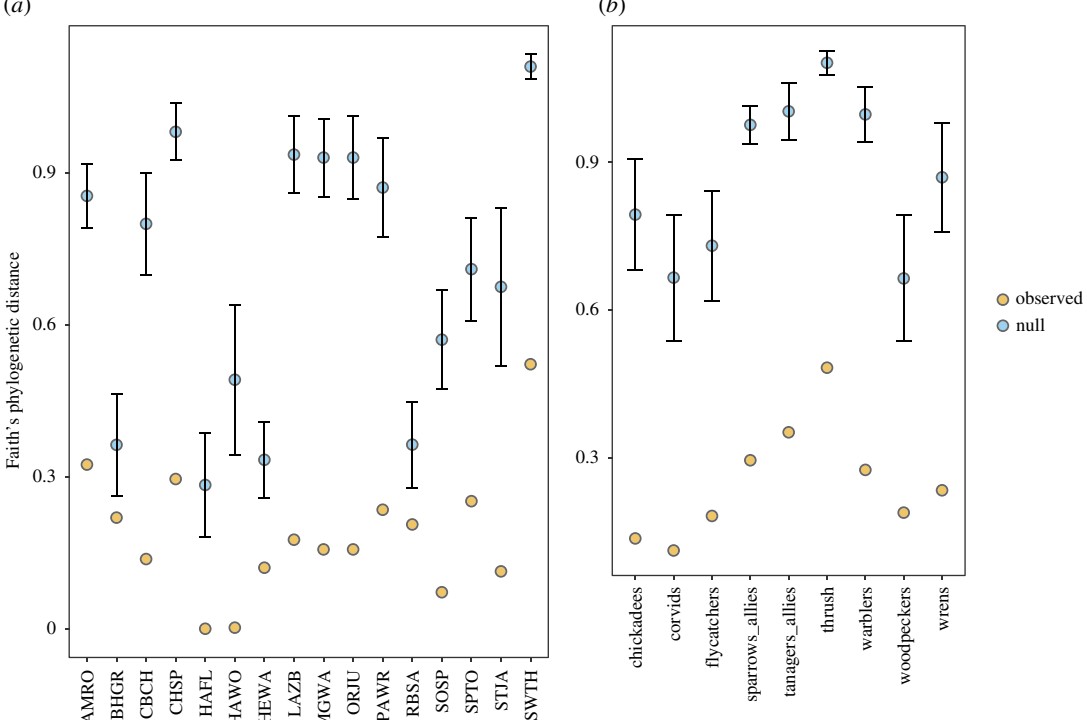

**Figure 4.** The observed (gold) phylogenetic distance of all of the bryophytes found on individual (*a*) species and (*b*) families of birds compared with null distribution means ± 95% CI (blue points) generated by placing bryophyte species randomly across the phylogenetic tree over 1000 iterations.

animals with similar traits rather than associations with specific species is the norm [83]. We therefore expect that with further work it may become apparent that individual bird species co-vector specific groups of bryophytes together. Thus, understanding these relationships will allow for better prediction of how the variety of birds in a given location may impact bryophyte dispersal outcomes.

Avian dispersal has the potential to vastly influence the dispersal of associated bryophyte species, especially with regards to directed dispersal to appropriate microhabitat. These associations with avian dispersers may influence the community and population structure of local bryophyte assemblages via species and gene flow between otherwise unconnected sites. These effects, in turn, may influence the functional role that bryophytes play in biotic interactions with bryophilic microbes and microfauna. In particular, the diversity generated and maintained by these processes contributes to a variety of the functional services provided by bryophytes, including creating microhabitats for an array of microarthropods, providing foraging grounds for insectivorous birds [84] and providing an appropriate source of nesting material. Thus, the avian contribution to bryophyte dispersal is likely to influence the resources available to birds themselves and could therefore impact both their behavioural ecology and fitness in a given locale. In order to understand the extent to which this transport mechanism may shape dispersal outcomes for bryophytes, we suggest that future work should focus on how propagule loads of various bird species may specifically influence the co-dispersal of bryophytes on bird surfaces to determine whether certain species pairs are more likely to arrive in novel sites together via an animal vector. Additionally, quantifying the topical retention time of spores and fragments on bird surfaces would allow us to predict the dispersal distance potential of bryophytes during both local and migratory avian movements.

Our investigation of bryophyte–bird networks expands the list of animals with the potential to disperse bryophytes to include a wide variety of passerines. The structure exhibited by this interaction network suggests that avian identity and foraging behaviour shape the identity and diversity of bryophytes carried by birds. Specifically, mid-canopy foragers are likely to carry the widest array of bryophytes, though the particular suite of bryophytes associated with a species varies within behavioural groups. Finally, despite these overall trends in bryophyte partner specificity in response to avian foraging guild, we note that particular species act idiosyncratically within our network, suggesting species-specific natural history.

Ethics. All birds were captured under U.S. Geological Survey Bird Banding permit number 22230 – S and were also overseen by the Portland State University Institute Animal Care and Use Committee (IACUC protocol no. 52).

Data accessibility. Data are available in the Dryad Digital Repository: https://doi.org/10.5061/dryad.1jwstqjs6 [85]. The code and data to reproduce the analysis are available at https://github.com/mwchmiel/windriver. Additionally, all bird capture data have been submitted to the U.S. Geological Survey's Bird Banding Laboratory.

Authors' contributions. M.W.C. designed the study with input from S.M.E. M.W.C. carried out the study and analysed the data. M.W.C. and S.M.E. wrote the manuscript. All authors gave final approval for publication and agreed to be held accountable for the work performed therein.

Competing interests. The authors have no competing interests to declare.

Funding. The Portland State Department of Biology Forbes Lea Research Fund, American Bryological and Lichenological Society Anderson and Crum Grant for Field Research in Bryology and National Science Foundation Doctoral Dissertation Improvement Grant (DEB-1701756) provided financial support.

Acknowledgements. The authors would like to thank Ariadna Covarrubias-Ornelas, Sara Herrejon Chavez and Samantha Martin for assistance in the laboratory. We would also like to thank Claudia Candia and Jess Shamek for assistance in the field. We thank the United States Forest Service, especially Ken Bible for logistical support in the field, and the Portland State University Institute Animal Care and Use Committee, Washington State Department of Fish and Wildlife and the United States Geological Survey Bird Banding Laboratory for permitting and animal research oversight. Finally, we would like to thank three anonymous reviewers who provided comments that have helped improve the manuscript.

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
