## [Peer Review File · Royal Society Open Science]

Review History

RSOS-201193.R0 (Original submission)

Review form: Reviewer 1

Is the manuscript scientifically sound in its present form?

Yes

Are the interpretations and conclusions justified by the results?

No

Is the language acceptable?

Yes

Do you have any ethical concerns with this paper?

No

Have you any concerns about statistical analyses in this paper?

Yes

Recommendation?

Major revision is needed (please make suggestions in comments)

Comments to the Author(s)

This is a study that follows up on a previous analysis of the same data set that was published in Proceeding B last year. I have three concerns regarding this manuscript:

- originality: objectives largely overlap with the 2019 paper. Although this new manuscript is focused on networks I found it too similar to the previous one and revolving around the same ideas. A symptom of this is that there is an entire paragraph in the Discussion that is an almost verbatim copy of one in the previous paper (page 14, line 49).

- hypotheses are too general. Network ecology has moved from merely describing networks to assessing the causes behind the patterns. In this sense, I suggest you think of more concise hypotheses regarding the association between bird traits and their role/degree of specialization in the network.

- specialization indices: H2 and d' measure selectivity, not specialization. I suggest you have a look at the help file of networklevel function in the bipartite R package, where this issue is discussed (not extensively though).

Relative abundances: the relative abundances of bird and bryophyte species may be constraining network structure. Perhaps compare the observed values against a null model that estimates the expected frequency of interaction based on the crossproduct of abundance vectors could deal with this issue (I suggest you have a look at Diego Vázquez and Carsten Dormann papers on this issue).

Review form: Reviewer 2

Is the manuscript scientifically sound in its present form?

Yes

Are the interpretations and conclusions justified by the results?

Yes

Is the language acceptable?

Yes

Do you have any ethical concerns with this paper?

No

Have you any concerns about statistical analyses in this paper?

No

Recommendation?

Accept with minor revision (please list in comments)

Comments to the Author(s)

This excellent and original manuscript falls within the scope of Royal Society Open Science and presents well-designed field study and analyses. This manuscript's focus is the bryophyte and bird species dispersal network, which could attract great interest in the field of dispersal ecology.

Overall the manuscript is well-written and fits well to the standards of the journal. I have only a few comments to the authors. Please see them below.

Methods:

With the cotton swabs, I assume you collected not just the spores, but some fragments of the bryophytes. If this is the case please correct.

How did you categorize the birds feeding guilds? What literature where used? For example, the hummingbird could be easily categorized by several guilds. Please add the literature, which was used.

It would be great to have a supplementary table with the birds and their traits (feeding behaviour).

page 6, line5 to 10: they can even survive the gut passage of waterbirds. See: Wilkinson et al. 2017 (Wilkinson, D. M., Lovas-Kiss, A., Callaghan, D. A., & Green, A. J. (2017). Endozoochory of large bryophyte fragments by waterbirds. *Cryptogamie, Bryologie*, 38(2), 223-228.)

page 7 line 53: „sampling“ do you mean sampled?

page 9 line 8: “phylogenetic” something missing here? distance?

page 14 line3: So if fragments easily, in the methods you only mention spores, but you could easily swab the fragments (which can be very viable) from the legs and feathers of the birds and not just spores right?

Decision letter (RSOS-201193.R0)

Dear Dr Chmielewski

The Editors assigned to your paper RSOS-201193 "Species-specific interactions in an avian-bryophyte dispersal network" have made a decision based on their reading of the paper and any comments received from reviewers.

Regrettably, in view of the reports received, the manuscript has been rejected in its current form. However, a new manuscript may be submitted which takes into consideration these comments.

We invite you to respond to the comments supplied below and prepare a resubmission of your manuscript. Below the referees' and Editors' comments (where applicable) we provide additional requirements. We provide guidance below to help you prepare your revision.

Please note that resubmitting your manuscript does not guarantee eventual acceptance, and we do not generally allow multiple rounds of revision and resubmission, so we urge you to make every effort to fully address all of the comments at this stage. If deemed necessary by the Editors, your manuscript will be sent back to one or more of the original reviewers for assessment. If the original reviewers are not available, we may invite new reviewers.

Please resubmit your revised manuscript and required files (see below) no later than 18-May-2021. Note: the ScholarOne system will 'lock' if resubmission is attempted on or after this

deadline. If you do not think you will be able to meet this deadline, please contact the editorial office immediately.

Please note article processing charges apply to papers accepted for publication in Royal Society Open Science (<https://royalsocietypublishing.org/rsos/charges>). Charges will also apply to papers transferred to the journal from other Royal Society Publishing journals, as well as papers submitted as part of our collaboration with the Royal Society of Chemistry (<https://royalsocietypublishing.org/rsos/chemistry>). Fee waivers are available but must be requested when you submit your manuscript (<https://royalsocietypublishing.org/rsos/waivers>).

Thank you for submitting your manuscript to Royal Society Open Science and we look forward to receiving your resubmission. If you have any questions at all, please do not hesitate to get in touch.

on behalf of Dr Sean Rands (Associate Editor) and Pete Smith (Subject Editor)
openscience@royalsociety.org

Associate Editor Comments to Author (Dr Sean Rands):

Associate Editor: 1

Comments to the Author:

Firstly, please accept my apologies for the extreme delay in getting a decision to you - I have been somewhat compromised by other duties associated with the ongoing pandemic, and the delay lies with me. I hope that you are all well and coping with the current conditions.

Thankyou for your submission, which I think presents some interesting biology - taking an editorial view as a generalist reader, I was intrigued by the natural history element of your manuscript, and think that conceptually it's a nice interesting piece. Your manuscript was independently read by two specialist reviewers, whose comments should be reproduced below. Based on their comments, which directed my own reading, I think that the manuscript would need some more analysis and some restructuring before it could be considered as suitable for publication, and it's not clear at this stage whether the changes would render the manuscript acceptable. I'm therefore suggesting this version is rejected, but that you have the opportunity to resubmit a new version if you wish.

This decision is based mostly on the criticisms of reviewer #1, who raises some concerns about the logic and methodology behind the network analyses. The following observations were also passed on, which I think are very relevant to what you would need to do in a revision. "The authors use two metrics (H2 and d') to describe network structure. In the manuscript these metrics are misinterpreted as measures of specialization when in fact they are measures of selectivity. While these apply well to animals, such as frugivores or pollinators, foraging on plants, their meaning when used to describe bird-bryophyte interactions is obscure."

"In addition, the departure from randomness of network structure was assessed using an unspecified null model (not available in github either). It is important that the authors describe

the model in detail because there are many possible null models and results are heavily contingent on the assumptions made.

Finally, the influence of the relative abundances of species seems to have been entirely overlooked when analyzing network patterns. In this context, claims such as "Our study system reflects a network structure in which certain bryophyte-bird associations are more common than others, and this demonstrates that bipartite species relationships show significant specialization" (line 55, page 12) lack a solid underpinning because some interactions may be more common than others simply because they involve two abundant species. A proper null model should include this neutral effect. In relation to this, the authors "measured the cover of bryophytes on both tree trunks and the ground by quadrat sampling every 10 meters along multiple transects". Such abundance data could be used to gauge bryophyte occurrence on birds' bodies."

Reviewer comments to Author:

Reviewer: 1

Comments to the Author(s)

This is a study that follows up on a previous analysis of the same data set that was published in Proceeding B last year. I have three concerns regarding this manuscript:

- originality: objectives largely overlap with the 2019 paper. Although this new manuscript is focused on networks I found it too similar to the previous one and revolving around the same ideas. A symptom of this is that there is an entire paragraph in the Discussion that is an almost verbatim copy of one in the previous paper (page 14, line 49).

- hypotheses are too general. Network ecology has moved from merely describing networks to assessing the causes behind the patterns. In this sense, I suggest you think of more concise hypotheses regarding the association between bird traits and their role/degree of specialization in the network.

- specialization indices: H_2 and d' measure selectivity, not specialization. I suggest you have a look at the help file of networklevel function in the bipartite R package, where this issue is discussed (not extensively though).

Relative abundances: the relative abundances of bird and bryophyte species may be constraining network structure. Perhaps compare the observed values against a null model that estimates the expected frequency of interaction based on the crossproduct of abundance vectors could deal with this issue (I suggest you have a look at Diego Vázquez and Carsten Dormann papers on this issue).

Reviewer: 2

Comments to the Author(s)

This excellent and original manuscript falls within the scope of Royal Society Open Science and presents well-designed field study and analyses. This manuscript's focus is the bryophyte and bird species dispersal network, which could attract great interest in the field of dispersal ecology. Overall the manuscript is well-written and fits well to the standards of the journal. I have only a few comments to the authors. Please see them below.

Methods:

With the cotton swabs, I assume you collected not just the spores, but some fragments of the bryophytes. If this is the case please correct.

How did you categorize the birds feeding guilds? What literature where used? For example, the hummingbird could be easily categorized by several guilds. Please add the literature, which was used.

It would be great to have a supplementary table with the birds and their traits (feeding behaviour).

page 6, line 5 to 10: they can even survive the gut passage of waterbirds. See: Wilkinson et al. 2017 (Wilkinson, D. M., Lovas-Kiss, A., Callaghan, D. A., & Green, A. J. (2017). Endozoochory of large bryophyte fragments by waterbirds. *Cryptogamie, Bryologie*, 38(2), 223-228.)

page 7 line 53: „sampling“ do you mean sampled?

page 9 line 8: “phylogenetic” something missing here? distance?

page 14 line 3: So if fragments easily, in the methods you only mention spores, but you could easily swab the fragments (which can be very viable) from the legs and feathers of the birds and not just spores right?

===PREPARING YOUR MANUSCRIPT===

===PREPARING YOUR REVISION IN SCHOLARONE===

Please ensure that you include a summary of your paper at Step 2 'Type, Title, & Abstract'. This should be no more than 100 words to explain to a non-scientific audience the key findings of your

research. This will be included in a weekly highlights email circulated by the Royal Society press office to national UK, international, and scientific news outlets to promote your work.

Author's Response to Decision Letter for (RSOS-201193.R0)

See Appendix A.

RSOS-211230.R0

Review form: Reviewer 2

Is the manuscript scientifically sound in its present form?

Yes

Are the interpretations and conclusions justified by the results?

Yes

Is the language acceptable?

Yes

Do you have any ethical concerns with this paper?

No

Have you any concerns about statistical analyses in this paper?

No

Recommendation?

Accept as is

Comments to the Author(s)

The authors have revised the manuscript taking into account almost all suggestions and comments. I have no further major issues regarding this manuscript. Regarding the first reviewer's comments on the statistics and use of terms, the authors tackled them and I suggest accepting the paper.

Review form: Reviewer 3

Is the manuscript scientifically sound in its present form?

Yes

Are the interpretations and conclusions justified by the results?

No

Is the language acceptable?

Yes

Do you have any ethical concerns with this paper?

No

Have you any concerns about statistical analyses in this paper?

No

Recommendation?

Major revision is needed (please make suggestions in comments)

Comments to the Author(s)

My main concern with this ms is overinterpretation of the results with exaggerated claims of the importance of the results (exemplified below). I suggest that the authors should focus on the natural history, which gives a story that is interesting enough.

Abstract. The first two sentences of abstract give the impression that there are very few studies on dispersal and community structure in spore plants. Yet several cases are cited, and the authors seem to have good knowledge of the literature. This is probably intended as a selling point but is not convincing. The same can be said about the statement that dispersal relationships between birds and bryophytes have never been examined.

4:24-36. This is one example where the text promises results that go beyond this study!

16:24. Here the importance for long-distance dispersal is suggested. Did you find diaspores of bryophytes that don't grow at the site or in the region. Otherwise this passage is too speculative.

17:15. The interpretations are stretched much too far. That these bird dispersal events are of importance for atmosphere-biosphere interactions and soil-atmosphere interface is not credible. Focus on the natural history and the relevance for the bird and moss communities (as later in the paragraph). This is interesting enough, and the conclusions would be within the scope of the results.

The final paragraph (17:33) is somewhat disappointing. It can make sense to mention potential future studies, but it would be nice to focus the final paragraphs on the main conclusions of the study.

Other comments

11:45. That the network is nonrandom is not surprising, but you could help the reader by picking out noteworthy observations from Fig. 1. What is it that makes the pattern non-random? For example the common mosses AUAN and ISMY seem to be strongly linked to the common bird SWTH, which seems as expected by random. In contrast, the most common moss (*Ceratodon*) is more linked to AMRO, than to SWTH, which seems non-random.

11:52 (text to Fig. 2). This was difficult to evaluate, because I did not fully understand Fig. 2 as the boxplot and brown vertical bar was not explained in legend. Conversely, the sentence "Specificity of the full network,..." should be in text, not in legend.

A technicality that I did not understand in Fig. 2: Is $\Delta H_2'$ the difference between the observed and the mean of the null distribution? Or the median (since the plot does not show mean)? Or perhaps I just misunderstood.

12:3. In the legend to Fig. 3 there are also some observations that should rather be in the text (eg. "While ground foraging tended to decrease specificity, specificity of associations for *Polytrichum juniperinum* increased. Ground foraging associations strongly drive overall network patterns", and others).

13:5, Fig. 4. Clarify here that it is phylogenetic distance of bryophytes. Again, the text should extract the interesting observations in the graphs. To just note that it was lower than expected seems trivial. What do you want the reader to see (which should then be dealt with in the discussion)?

13:18. The statement that this is the "first species-specific interaction network between birds and topically resident bryophyte propagules" seems to have overlooked the paper by Fonturbel (Functional Ecology, 35 (2021), 226-238).

Decision letter (RSOS-211230.R0)

Dear Dr Chmielewski

The Editors assigned to your paper RSOS-211230 "Species-specific interactions in an avian-bryophyte dispersal network" have now received comments from reviewers and would like you to revise the paper in accordance with the reviewer comments and any comments from the Editors. Please note this decision does not guarantee eventual acceptance.

Please submit your revised manuscript and required files (see below) no later than 21 days from today's (ie 30-Sep-2021) date. Note: the ScholarOne system will 'lock' if submission of the revision is attempted 21 or more days after the deadline. If you do not think you will be able to meet this deadline please contact the editorial office immediately.

on behalf of Dr Sean Rands (Associate Editor) and Pete Smith (Subject Editor)
openscience@royalsociety.org

Reviewer comments to Author:

Reviewer: 3

Comments to the Author(s)

My main concern with this ms is overinterpretation of the results with exaggerated claims of the importance of the results (exemplified below). I suggest that the authors should focus on the natural history, which gives a story that is interesting enough.

Abstract. The first two sentences of abstract give the impression that there are very few studies on dispersal and community structure in spore plants. Yet several cases are cited, and the authors seem to have good knowledge of the literature. This is probably intended as a selling point but is not convincing. The same can be said about the statement that dispersal relationships between birds and bryophytes have never been examined.

4:24-36. This is one example where the text promises results that go beyond this study!

16:24. Here the importance for long-distance dispersal is suggested. Did you find diaspores of bryophytes that don't grow at the site or in the region. Otherwise this passage is too speculative.

17:15. The interpretations are stretched much too far. That these bird dispersal events are of importance for atmosphere-biosphere interactions and soil-atmosphere interface is not credible. Focus on the natural history and the relevance for the bird and moss communities (as later in the paragraph). This is interesting enough, and the conclusions would be within the scope of the results.

The final paragraph (17:33) is somewhat disappointing. It can make sense to mention potential future studies, but it would be nice to focus the final paragraphs on the main conclusions of the study.

Other comments

11:45. That the network is nonrandom is not surprising, but you could help the reader by picking out noteworthy observations from Fig. 1. What is it that makes the pattern non-random? For example the common mosses AUAN and ISMY seem to be strongly linked to the common bird SWTH, which seems as expected by random. In contrast, the most common moss (*Ceratodon*) is more linked to AMRO, than to SWTH, which seems non-random.

11:52 (text to Fig. 2). This was difficult to evaluate, because I did not fully understand Fig. 2 as the boxplot and brown vertical bar was not explained in legend. Conversely, the sentence "Specificity of the full network,..." should be in text, not in legend.

A technicality that I did not understand in Fig. 2: Is $\Delta H_2'$ the difference between the observed and the mean of the null distribution? Or the median (since the plot does not show mean)? Or perhaps I just misunderstood.

12:3. In the legend to Fig. 3 there are also some observations that should rather be in the text (eg. "While ground foraging tended to decrease specificity, specificity of associations for *Polytrichum juniperinum* increased. Ground foraging associations strongly drive overall network patterns", and others).

13:5, Fig. 4. Clarify here that it is phylogenetic distance of bryophytes. Again, the text should extract the interesting observations in the graphs. To just note that it was lower than expected seems trivial. What do you want the reader to see (which should then be dealt with in the discussion)?

13:18. The statement that this is the "first species-specific interaction network between birds and topically resident bryophyte propagules" seems to have overlooked the paper by Fonturbel (Functional Ecology, 35 (2021), 226-238).

Reviewer: 2

Comments to the Author(s)

The authors have revised the manuscript taking into account almost all suggestions and comments. I have no further major issues regarding this manuscript. Regarding the first reviewer's comments on the statistics and use of terms, the authors tackled them and I suggest accepting the paper.

===PREPARING YOUR MANUSCRIPT===

===PREPARING YOUR REVISION IN SCHOLARONE===

Author's Response to Decision Letter for (RSOS-211230.R0)

See Appendix B.

RSOS-211230.R1

Review form: Reviewer 3

Is the manuscript scientifically sound in its present form?

Yes

Are the interpretations and conclusions justified by the results?

Yes

Is the language acceptable?

Yes

Do you have any ethical concerns with this paper?

No

Have you any concerns about statistical analyses in this paper?

No

Recommendation?

Accept as is

Comments to the Author(s)

Much improved, and I am pleased with the better focus in this version.

Decision letter (RSOS-211230.R1)

Dear Dr Chmielewski,

It is a pleasure to accept your manuscript entitled "Species-specific interactions in an avian-bryophyte dispersal network" in its current form for publication in Royal Society Open Science.

The comments of the reviewer(s) who reviewed your manuscript are included at the foot of this letter.

on behalf of Professor Pete Smith (Subject Editor)
openscience@royalsociety.org

Associate Editor Comments to Author:

Congratulations on the paper - it is now ready for acceptance. Thank you for supporting RSOS.

Reviewer comments to Author:

Reviewer: 3

Comments to the Author(s)

Much improved, and I am pleased with the better focus in this version.

Appendix A

Dear Dr. Sean Rands and the Royal Society Open Science editorial team,

We have taken the great suggestions by the AE and reviewers below, and think it has marked improved our manuscript. In order to address the specific comments, we provide a point-by-point response to the original comments below.

Thank you for your time and patience,

Matthew Chmielewski, Ph.D.

Manuscript ID RSOS-201193: Response to referees and editors

Below, we provide line-by-line responses to the reviewer commentary along with editorial concerns. We think that the manuscript, having integrated these suggestions, has been markedly improved.

Associate Editor Comments to Author (Dr Sean Rands):

Associate Editor: 1

Comments to the Author:

Firstly, please accept my apologies for the extreme delay in getting a decision to you - I have been somewhat compromised by other duties associated with the ongoing pandemic, and the delay lies with me. I hope that you are all well and coping with the current conditions.

Thank you for your submission, which I think presents some interesting biology - taking an editorial view as a generalist reader, I was intrigued by the natural history element of your manuscript, and think that conceptually it's a nice interesting piece.

Your manuscript was independently read by two specialist reviewers, whose comments should be reproduced below. Based on their comments, which directed my own reading, I think that the manuscript would need some more analysis and some restructuring before it could be considered as suitable for publication, and it's not clear at this stage whether the changes would render the manuscript acceptable. I'm therefore suggesting this version is rejected, but that you have the opportunity to resubmit a new version if you wish.

This decision is based mostly on the criticisms of reviewer #1, who raises some concerns about the logic and methodology behind the network analyses. The following observations were also passed on, which I think are very relevant to what you would need to do in a revision. "The authors use two metrics (H2 and d') to describe network structure. In the manuscript these metrics are misinterpreted as measures of specialization when in fact they are measures of selectivity. While these apply well to

animals, such as frugivores or pollinators, foraging on plants, their meaning when used to describe bird-bryophyte interactions is obscure."

We discuss the concerns regarding network metric choice below in response to the comments provided by reviewer 1.

"In addition, the departure from randomness of network structure was assessed using an unspecified null model (not available in github either). It is important that the authors describe the model in detail because there are many possible null models and results are heavily contingent on the assumptions made.

We agree that the null model specifications are important in interpreting our approach and that we overlooked making this clear and available via github. We now both explicitly address the null model specifications in the text, and have made the code available on github.

Finally, the influence of the relative abundances of species seems to have been entirely overlooked when analyzing network patterns. In this context, claims such as "Our study system reflects a network structure in which certain bryophyte-bird associations are more common than others, and this demonstrates that bipartite species relationships show significant specialization" (line 55, page 12) lack a solid underpinning because some interactions may be more common than others simply because they involve two abundant species. A proper null model should include this neutral effect.

This concern is related to the null model approach and is articulated as a concern by reviewer 1 below. We address the null model specifications relative to reviewer 1's concerns below.

In relation to this, the authors "measured the cover of bryophytes on both tree trunks and the ground by quadrat sampling every 10 meters along multiple transects". Such abundance data could be used to gauge bryophyte occurrence on birds' bodies."

Reviewer comments to Author:

Reviewer: 1

Comments to the Author(s)

This is a study that follows up on a previous analysis of the same data set that was published in Proceeding B last year. I have three concerns regarding this manuscript: - originality: objectives largely overlap with the 2019 paper. Although this new manuscript is focused on networks I found it too similar to the previous one and revolting around the same ideas. A symptom of this is that there is an entire paragraph in the Discussion that is an almost verbatim copy of one in the previous paper (page 14, line 49).

We have edited the text to reflect an emphasis on how this particular manuscript is novel compared to the previous work. We maintain that describing species associations within the network is an entirely unique product from these data than previously studied topical spore loads, and thus merits publication.

- hypotheses are too general. Network ecology has moved from merely describing networks to assessing the causes behind the patterns. In this sense, I suggest you think of more concise hypotheses regarding the association between bird traits and their role/degree of specialization in the network.

We address this concern by providing more explicit predictions within the paper linked to our analyses, with new interpretations.

- specialization indices: H_2 and d' measure selectivity, not specialization. I suggest you have a look at the help file of networklevel function in the bipartite R package, where this issue is discussed (not extensively though).

Bluthen, Menzel, and Bluthgen introduce the derivation of d' and H_2' in Bluthgen *et al.* 2006 a paper entitled Measuring specialization in species interaction networks. Thus, they refer to these as measures of specialization from the onset of the title of their paper.

Specifically, they introduce the species-level index (d') as a measure of “partner diversity”, derived from information theory formalized as Kullback-Leibler distances. They go on to refer to both d' and H_2' as specialization, e.g. in the passage:

“Our study suggests that d' and H_2' represent scale-independent and meaningful indices to characterize specialization on the level of single species and the entire network, respectively.” (Bluthgen 2006)

and in 2011:

“Quantitative specialization indices d' and H_2' [43]. The index d' (species-level specialization) describes the exclusiveness of a species, i.e. its quantitative deviation from the overall distribution of all bees on resin sources or of the overall distribution of compounds on all bees. The related network-level specialization index H_2' characterizes the overall quantitative partitioning of resin sources or chemical compounds across species. Both measures range between 0 (each species uses the same resin sources or has identical chemistry) and 1 (species uses a different set of resins or have unique compounds, i.e. complementary specialization).” (Leonhardt, Schmitt, and Bluthgen 2011)

We therefore used the term “specialization” in its original context as described by Bluthgen *et al.* rather than selectivity.

We additionally examined the networklevel function of the bipartite package documentation, in which Dormann refers to these metrics with the term “specialization”.

Nico Bluthgen has gone on to refer to these indices as “selectivity”, “complementary specialization”, “exclusiveness”, and “specificity”.

We agree with Reviewer 2 that the use of these various specific terms can lead to a textual interpretation of the underlying meaning of these indices that can vary depending on the term used.

The dominant application of these methods to describe plant-pollinator and plant-frugivore networks may have lead to a dominant use of “selectivity”, with the interpretation being that animals are making choices based on tightly co-evolved interactions. Despite this, these indices are derived ultimately from matrix mathematics in which the topology of nodes and edges of a network are being described. The emphasis on animal choice of resources is therefore an interpretation of the network structure, rather than being inherent to any interpretation of these indices.

Indeed, these indices have gone on to be used by Bluthgen and others to describe various networks in which co-evolved “specialization” or behavioral choice of resources (“selectivity”) may not be the most appropriate ecological description of the studied relationship.

These include studies involving plant-microbe interactions, epiphyte-phorophyte commensalisms, insect-chemical network/tree-chemical network, and mite-habitat associations.

Following these examples, we interpret the patterns of bird-bryophyte interaction in our system as a product of avian use of particular niche space throughout the habitat which differentially exposes them to particular bryophytes.

We agree again with Reviewer 2 that the terms “specialization” and “selectivity” can be interpreted to imply the coevolution of a dispersal syndrome in which particular birds are choosing particular mosses. While this has been shown to be true for avian choice of nesting material (and likely a factor in our system), and may be true for avian use of foraging substrates, we don’t make the claim that we can interpret this to be true within our study system. We therefore have chosen to change out language to move away from “specialization” and instead to “specificity” of interaction.

While we feel that this helps disambiguate our meaning, we add additional text to the methods to clarify the specific meaning of our use of these indices by discussing the derivation as well as providing examples that align with our use of these terms.

Relative abundances: the relative abundances of bird and bryophyte species may be constraining network structure. Perhaps compare the observed values against a null model that estimates the expected frequency of interaction based on the crossproduct of abundance vectors could deal with this issue (I suggest you have a look at Diego Vázquez and Carsten Dormann papers on this issue).

We agree that both asymmetric sampling and null model selection can have profound impacts on the outcome of bipartite network indices. Nico Bluthgen, indeed, argues that generally measures of partner interactions increase with sampling effort, but that d' and H^2 are specifically built to account for asymmetric sampling:

“In interaction datasets that are limited by sampling, however, partner diversity simply increases with the number of observations, whereas selectivity (e.g. H^2 , d') can be determined independently of the completeness of interactions observed.” Kaiser-Bunbury and Bluthgen (2015)

Regardless, Reviewer 1 is correct in criticizing an approach employing an undisclosed null model. This was an oversight that we have now corrected, including the null model in both the text and the available code.

As suggested by Reviewer 1, we employ Dormann’s null.dist function which fits a distribution to the observed data, then generates null webs based on the crossproduct of abundance vectors from these distributions. We discuss this explicitly in the methods, improving the clarify of our approach.

Reviewer: 2

Comments to the Author(s)

This excellent and original manuscript falls within the scope of Royal Society Open Science and presents well-designed field study and analyses. This manuscript’s focus is the bryophyte and bird species dispersal network, which could attract great interest in the field of dispersal ecology. Overall the manuscript is well-written and fits well to the standards of the journal. I have only a few comments to the authors. Please see them below.

Methods:

With the cotton swabs, I assume you collected not just the spores, but some fragments of the bryophytes. If this is the case please correct.

We have gone back and made it apparent that we are discussing bryophyte propagules rather than spores per se. We find the distinction between seed dispersal and dispersal of spore-bearing plants to be important to our introduction, so have kept our focus on spore-bearing plants in place. We think that with the additional language we have made it clear that we are not sampling only spores for this study.

How did you categorize the birds feeding guilds? What literature where used? For example, the hummingbird could be easily categorized by several guilds. Please add the literature, which was used.

The previous approach used the Cornell Lab of Ornithology's Birds of North America resource in order to classify feeding guilds.

When considering a response to this review, we decided to move beyond simply citing those resources and have changed to a more quantitative approach of classification based on the Elton Traits database.

We believe that this approach is a) more explicitly repeatable, b) classifies species based on a quantitative threshold, and c) allows species to belong to multiple guilds, better reflecting the complexity of resource us by various birds.

We explain this approach in the revised text.

It would be great to have a supplementary table with the birds and their traits (feeding behaviour).

We have generated our new traits from a published database, clarifying this point.

page 6, line5 to 10: they can even survive the gut passage of waterbirds. See: Wilkinson et al. 2017 (Wilkinson, D. M., Lovas-Kiss, A., Callaghan, D. A., & Green, A. J. (2017). Endozoochory of large bryophyte fragments by waterbirds. *Cryptogamie, Bryologie*, 38(2), 223-228.)

Indeed! We are specifically focused on epizoochory but mentioned synzoochory via hummingbirds and think it is appropriate to also mentioned endozoochory so have added a reference here. Thanks!

page 7 line 53: „sampling” do you mean sampled?

Corrected

page 9 line 8: “phylogenetic” something missing here? distance?

Corrected

page 14 line3: So if fragments easily, in the methods you only mention spores, but you

could easily swab the fragments (which can be very viable) from the legs and feathers of the birds and not just spores right?

We have gone back and changed the methods relate to swab sampling to use the general term “propagules”

===PREPARING YOUR MANUSCRIPT===

- one version identifying all the changes that have been made (for instance, in coloured highlight, in bold text, or tracked changes);
- a 'clean' version of the new manuscript that incorporates the changes made, but does not highlight them. This version will be used for typesetting if your manuscript is accepted.

Appendix B

Dear Royal Society Open Science editorial staff,

Below we provide a point-by-point response to the editorial and reviewer comments on manuscript RSOS-211230.

Response to comments from the editor:

This review came without additional editorial comments, which we took to mean that the editorial concerns were sufficiently outlined by the reviewers during this round.

Response to comments from reviewers:

Reviewer comments to Author:

Reviewer: 3

Comments to the Author(s)

My main concern with this ms is overinterpretation of the results with exaggerated claims of the importance of the results (exemplified below). I suggest that the authors should focus on the natural history, which gives a story that is interesting enough.

We appreciate the suggestions to temper our language surrounding the scope of the findings. We have removed these overextensions throughout the manuscript following this and additional suggestions below, and have added specifics regarding the relevant natural history findings.

Abstract. The first two sentences of abstract give the impression that there are very few studies on dispersal and community structure in spore plants. Yet several cases are cited, and the authors seem to have good knowledge of the literature. This is probably intended as a selling point but is not convincing. The same can be said about the statement that dispersal relationships between birds and bryophytes have never been examined.

We agree that the wording missed the mark here. While we maintain that ours is the first study to examine this phenomenon across a wide range of passerines, we have adjusted the language here and throughout to better reflect the role that our study plays in the literature.

In the abstract here we specifically, we frame the spore dispersal by animals literature as lacking relative to the seed dispersal literature.

We adjust the second concern by outlining that we know little about species-specific patterns of association between passerine birds and epizoochory.

4:24-36. This is one example where the text promises results that go beyond this study!

We agree that these suggestions are not particularly relevant to the current study and have removed them.

16:24. Here the importance for long-distance dispersal is suggested. Did you find diaspores of bryophytes that don't grow at the site or in the region. Otherwise this passage is too speculative.

We changed our language here to make sure that it is clear that Lewis *et al.* have suggested that LDD by birds may be important, not that we have direct evidence for LDD in this study.

17:15. The interpretations are stretched much too far. That these bird dispersal events are of importance for atmosphere-biosphere interactions and soil-atmosphere interface is not credible. Focus on the natural history and the relevance for the bird and moss communities (as later in the paragraph). This is interesting enough, and the conclusions would be within the scope of the results.

We agree that while all of these roles of bryophytes are important, they are only tangentially linked to our study and we have therefore removed them.

The final paragraph (17:33) is somewhat disappointing. It can make sense to mention potential future studies, but it would be nice to focus the final paragraphs on the main conclusions of the study.

This is a good point and we have revised the manuscript to a) combine future directions with the previous paragraph and b) end with a better summation of the findings of the study.

Other comments

11:45. That the network is nonrandom is not surprising, but you could help the reader by picking out noteworthy observations from Fig. 1. What is it that makes the pattern non-random? For example the common mosses AUAN and ISMY seem to be strongly linked to the common bird SWTH, which seems as expected by random. In contrast, the most common moss (Ceratodon) is more linked to AMRO, than to SWTH, which seems non-random.

We agree that discussing some of the specifics of the network as presented in Figure 1 help guide the reader toward the additional findings. We discuss some specifics of the network as suggested here.

11:52 (text to Fig. 2). This was difficult to evaluate, because I did not fully understand Fig. 2 as the boxplot and brown vertical bar was not explained in legend. Conversely, the sentence "Specificity of the full network,..." should be in text, not in legend.

We have reworded the figure legend to make it more clear what the vertical bars and green boxplots specify. The text "Specificity of the full network..." is redundant with finding reported in the text and we have clarified some portions on this in the text.

A technicality that I did not understand in Fig. 2: Is $\Delta H_2'$ the difference between the observed

and the mean of the null distribution? Or the median (since the plot does not show mean)?. Or perhaps I just misunderstood.

In clarifying the above, we hope we have made it clear that we are displaying the observed values of H_2' and the null distributions as boxplots, and that we provide statistical tests relative to $\Delta H_2'$ in the text.

12:3. In the legend to Fig. 3 there are also some observations that should rather be in the text (eg. "While ground foraging tended to decrease specificity, specificity of associations for *Polytrichum juniperinum* increased. Ground foraging associations strongly drive overall network patterns", and others).

We removed observations as suggested, some of which were in the text already.

13:5, Fig. 4. Clarify here that it is phylogenetic distance of bryophytes. Again, the text should extract the interesting observations in the graphs. To just note that it was lower than expected seems trivial. What do you want the reader to see (which should then be dealt with in the discussion)?

We originally stated "Faith's phylogenetic distance of all bryophytes..." but have reworded slightly to emphasize that we are talking about bryophytes

13:18. The statement that this is the "first species-specific interaction network between birds and topically resident bryophyte propagules" seems to have overlooked the paper by Fonturbel (Functional Ecology, 35 (2021), 226-238).

We respectfully disagree with this characterization of the specifics that make our findings distinct relative to Fonturbel 2021 and other work that we cite.

By species-specific here we intended to convey that our study is the first to include multiple bird species and their known bryophyte associates. Fonturbel 2021 includes only one bird species, *Sephanoides sephaniodes* and involves synzoochory rather than topically carried propagules (epizoochory).

We reword our statement here to more clearly convey this distinction, as well as the distinction that our work is the first work like this that involves passerines (*Sephanoides sephaniodes* being in the Apodiformes, and other studies we cite involves waterbirds and shorebirds and these do not detail the range of bryophyte species involved). We reword this to make our meaning clearer. As Fonturbel 2021 is a relevant citation, we instead include a reference to this work in the introduction.

Reviewer: 2

Comments to the Author(s)

The authors have revised the manuscript taking into account almost all suggestions and comments. I have no further major issues regarding this manuscript. Regarding the first

reviewer's comments on the statistics and use of terms, the authors tackled them and I suggest accepting the paper.

Thanks reviewer 2!